# Chemical Compounds Released from Specific Osteoinductive Bioactive Materials Stimulate Human Bone Marrow Mesenchymal Stem Cell Migration

**DOI:** 10.3390/ijms23052598

**Published:** 2022-02-26

**Authors:** Krzysztof Łukowicz, Barbara Zagrajczuk, Karolina Truchan, Łukasz Niedzwiedzki, Katarzyna Cholewa-Kowalska, Anna M. Osyczka

**Affiliations:** 1Department Biology and Cell Imaging, Institute of Zoology and Biomedical Research, Faculty of Biology, Jagiellonian University, Gronostajowa 9, 30-387 Krakow, Poland; krzysztof.lukowicz@doctoral.uj.edu.pl (K.Ł.); karolina.truchan@doctoral.uj.edu.pl (K.T.); 2Department of Glass Technology and Amorphous Coatings, Faculty of Materials Science and Ceramics, AGH University of Science and Technology, Mickiewicza Ave. 30, 30-059 Krakow, Poland; zagrab@agh.edu.pl (B.Z.); cholewa@agh.edu.pl (K.C.-K.); 3Department of Orthopedics and Physiotherapy, Faculty of Health Sciences, Jagiellonian University Medical College, Kopernika 19e, 31-501 Krakow, Poland; lukasz.niedzwiedzki@uj.edu.pl

**Keywords:** composites, stem cells, cell migration

## Abstract

In this work, a poly(L-lactide-*co*-glycolide) (PLGA)-based composite was enriched with one of the following sol-gel bioactive glasses (SBG) at 50 wt.%: A1—40 mol% SiO_2_, 60 mol% CaO, CaO/SiO_2_ ratio of 1.50; S1—80 mol% SiO_2_, 20 mol% CaO, CaO/SiO_2_ ratio of 0.25; A2—40 mol% SiO_2_, 54 mol% CaO, 6 mol% P_2_O_5_, CaO/SiO_2_ ratio of 1.35; S2—80 mol% SiO_2_,16 mol% CaO, 4 mol% P_2_O_5_, CaO/SiO_2_ ratio of 0.20. The composites and PLGA control sheets were then soaked for 24 h in culture media, and the obtained condition media (CM) were used to treat human bone marrow stromal cells (hBMSCs) for 72 h. All CMs from the composites increased ERK 1/2 activity vs. the control PLGA CM. However, expressions of cell migration-related c-Fos, osteopontin, matrix metalloproteinase-2, C-X-C chemokine receptor type 4, vascular endothelial growth factor, and bone morphogenetic protein 2 were significantly increased only in cells treated with the CM from the A1/PLGA composite. This CM also significantly increased the rate of human BMSC migration but did not affect cell metabolic activity. These results indicate important biological markers that are upregulated by products released from the bioactive composites of a specific chemical composition, which may eventually prompt osteoprogenitor cells to colonize the bioactive material and accelerate the process of tissue regeneration.

## 1. Introduction

During natural bone remodeling or its repair as a result of fractures, diseases, or injuries, mesenchymal stem cells (MSCs) migrate to the site of injury and differentiate into osteoblasts to regenerate bone tissue. Hence, several attempts have been made to use MSCs for the treatment of bone defects [1,2]. So far, most bone-related therapies focus on promoting MSC differentiation into osteoblasts. However, the migration of mesenchymal cells to the bone regeneration site is also a key issue in bone formation and repair. Incorrect MSC migration can result in several diseases, difficulties in treating fractures, as well as imbalances in osteoblastogenesis and osteoclastogenesis, resulting in bone resorption rather than bone formation [1]. 

Recent studies have stressed the issue of obtaining the correct MSC migration to the site of bone tissue repair to increase the effectiveness of any potential MSC-based bone therapy [1]. Bioactive materials are frequently used in orthopedics, maxillofacial surgery, and bone tissue engineering. Although “bioactivity” is a broad term, certain bioactive materials, upon contact with body fluids, release calcium ions, oxyanionic phosphate, and silicate species from their surface, which may play a role in the ossification process. Calcium ions are believed to be primary factors in increasing cell mobility, but given the different rates of calcium release from different bioactive materials, as well as the other ions that are also released, the overall effects of such material surface activity on cell migration may be distinct [3,4,5]. In this work, we examined poly(L-lactide-*co*-glycolide) (PLGA)-based composites containing gel-derived bioactive glasses (SBG) from either the SiO_2_–CaO system (namely A1—40 mol% SiO_2_, 60 mol% CaO, CaO/SiO_2_ ratio of 1.50; S1—80 mol% SiO_2_, 20 mol% CaO, CaO/SiO_2_ ratio of 0.25) or the SiO_2_–CaO–P_2_O_5_ system (i.e., A2—40 mol% SiO_2_, 54 mol% CaO, 6 mol% P_2_O_5_, CaO/SiO_2_ ratio of 1.35; S2—80 mol% SiO_2_, 16 mol% CaO, 4 mol% P_2_O_5_, CaO/SiO_2_ ratio of 0.20). SBGs were incorporated into the PLGA matrix at 50 wt.%. The studies regarding these materials have focused so far on direct cell–material interactions, but the products released from these materials may also play a biological function, especially due to their high bioactivity and osteoinductivity [6,7]. 

In this study, we tested the hypothesis that the products released from SBG/PLGA bioactive composites stimulate BMSC migration. The reason behind this is that before bone is formed, osteoblast precursor cells need to be attracted first to the site of bone regeneration to start the process of bone formation, and products released from the studied composites may contribute to this process. Despite that the main bioactive components of the studied composites are SBGs, the clinical applications of bioactive glasses on their own are limited, whereas their use as composite components is very broad and the type of matrix they are incorporated into influences their bioactivity. Thus, it was possible to examine them as components of PLGA-based composites. To verify the effects of hydrogenated compounds from these bioactive composites, they were preincubated in culture media to obtain “condition media” (CM), which were then used to treat human BMSC cultures. In addition to a typical cell migration study, the cellular metabolic activity, activation of the ERK1/2 pathway, and the expression of selected genes, namely c-Fos, osteopontin (OPN), matrix metalloproteinase-2 (MMP-2), C-X-C chemokine receptor type 4 (CXCR4), vascular endothelial growth factor (VEGF), and bone morphogenetic protein 2 (BMP-2), related to the cell migration process and/or osteogenesis, were examined. 

## 2. Results

The bioactivity of materials can be determined by the chemical composition of the base, the chemical groups exposed to the biological environment, and by ions or other material degradation products released from the material to the physiological environment [8,9]. This study was focused on the selected chemical compounds released from the obtained SBG/PLGA composites. Previous studies with these composites have shown that they rapidly release calcium ions to cell culture media upon an initial 24-h incubation period. Moreover, the levels of Ca released from these composites in a 24 h incubation time depend on the content of calcium in the SBG/PLGA composites. This could be explained by the chemical composition of SBGs incorporated into the PLGA, that is, the calcium release decreased as follows: A1 > A2 > S1 > S2 (Table 1) [7]. 

Given the highest amounts of calcium released into the culture medium by the A1/PLGA composites, this corresponded well with the gene expression analyses performed in this work (Figure 1). Human BMSCs exposed for 72 h to the CM from the A1/PLGA had significantly elevated c-Fos, OPN, MMP-2, VEGF, and CXCR4 mRNA levels, the transcripts potentially involved in the cell migration process. However, the overall metabolic activity of cells was comparable for all studied CMs from the composites and comparable to the PLGA CM control. 

Notably, compared to the PLGA control, human BMSCs treated with the CM from the A1/PLGA displayed the highest BMP-2 expression (Figure 2). Although the CM from the S2/PLGA composites also elevated BMP-2 expression, the BMP-2 mRNA levels in the hBMSCs treated with the CM from the S2/PLGA were significantly lower than in cells treated with the CM from the A1/PLGA. The CM from the A1/PLGA also significantly increased the cell migration rate, as measured by a QCM Chemotaxis Cell Migration Assay. Interestingly, the ERK 1/2 activity was significantly higher in cells treated with all studied composite-derived CMs vs. the CM from PLGA, but again, the highest ERK 1/2 activity was detected for the CM from the A1/PLGA composites. 

## 3. Discussion

There are several different molecules and processes involved in cell migration [10,11]. In the present study, the expressions of BMP-2, MMP-2, OPN, VEGF, and c-Fos were examined due to their potential involvement in the human BMSC initial response to the chemical compounds released from the studied materials [10]. The products of these genes are involved in processes such as bone regeneration and wound healing, which involve the mechanisms of the migration of cells toward the site of tissue regeneration [11,12,13]. On top of the above, we examined CXCR4 expression as one of the major factors controlling the processes of migration and the homing of MSCs [11,14,15,16]. At the time of implantation, the bioactive material is immediately exposed to the action of physiological fluids, and it begins to release active compounds to the immediate environment. It is well established that the active compounds released from such materials may initiate a biological response [17]. Given that the composites studied in this work contained gel-derived bioactive glasses of specific chemical compositions and released substantial amounts of Ca, Si, and P ions during the first 24-h contact with the physiological fluids (Table 1), it was possible to define whether the chemical products released by these materials are chemoattractants to ostoprogenitor cells that contribute on their own to the initiation of osteogenesis. In this work, calcium ions released from the studied surfaces were assumed to be the key ions responsible for the initiation of hBMSC migration toward the bioactive material. 

As shown in Table 1, all composites studied here released calcium ions, but the highest Ca levels were detected in the culture media harvested from the high-calcium A1/PLGA material [7]. Calcium ions have been shown to increase the expressions of BMP-2, CXCR4, OPN, c-Fos, VEGF, and MMP-2, and this is consistent with the results of the present study, where the CM from the A1/PLGA significantly increased the mRNA levels of the above-mentioned genes in human BMSC [15,17,18,19,20,21,22,23,24,25]. Notably, the expression of the above-mentioned genes decreased significantly for the A2/PLGA composites vs. those of the A1/PLGA. This can be partly explained by the reduced amounts of calcium ions released by the A2/PLGA composites. The materials enriched with P_2_O_5_ can quickly bind the calcium ions, accelerating the deposition of calcium phosphates on the composite surfaces, thus reducing the biological activity of free calcium ions [25]. Furthermore, the correlation between the VEGF expression and phosphorus oxide-containing materials (A2/PLGA and S2/PLGA) could be observed, as CMs from these materials reduced the mRNA levels of VEGF expression, which is consistent with reports showing that phosphorus compounds can affect VEGF expression [26]. A similar trend could be observed in the case of MMP-2. Phosphorus compounds are used as inhibitors of metalloproteinases [27]. Moreover, OPN expression correlated with the relative concentrations of calcium ions to phosphate ions (Ca^2+^/PO_4_^3−^ ratio), and calcium ions to silicate ions (Ca^2+^/SiO_4_^4−^ ratio). Namely, the CM collected from the A1/PLGA (Ca^2+^/PO_4_^3−^—17.01, Ca^2+^/SiO_4_^4−^—7.90) exhibited both Ca^2+^/PO_4_^3−^ and Ca^2+^/SiO_4_^4−^ ratios higher than the medium collected from the A2/PLGA (Ca^2+^/PO_4_^3−^—11.89, Ca^2+^/SiO_4_^4−^—4.60). Similarly, for the composites with S-type bioglasses, these parameters were higher for the S1/PLGA (Ca^2+^/PO_4_^3−^—4.67, Ca^2+^/SiO_4_^4−^—8.73) than for the S2/PLGA (Ca^2+^/PO_4_^3−^—2.12, Ca^2+^/SiO_4_^4−^—2.05). Assuming that calcium ions can stimulate OPN expression while silicate ions at higher concentrations can reduce OPN expression, and that media from the A1/PLGA and S1/PLGA showed elevated relative concentrations of calcium ions, it can be assumed that calcium ions were mainly responsible for the stimulation of OPN expression [13,28].

Furthermore, all CMs harvested from the studied composites increased the ERK 1/2 activity in human BMSCs vs. the CM from the control PLGA, which can have several cellular consequences. It has been reported that silica exposure activates the ERK 1/2 pathway in different types of cells, and the presence of silicon dioxide in the studied composites may be responsible for this effect [29,30,31]. However, the activation of ERK 1/2 by silicon compounds may be related to biological processes other than cell migration, as silicon compounds have been shown to inhibit cell mobility [32,33]. This was most visible when comparing the effects of CMs from the A1/PLGA and S2/PLGA composites. Although increased ERK 1/2 activity may also contribute to increased cell mobility, in this study, the hBMSC mobility was significantly increased only for the CM harvested from the A1/PLGA [34]. Thus, the other factors are necessary to mobilize hBMSC upon stimulation with CMs from composites other than A1/PLGA. The lack of significant increases in the migration rates of cells exposed to CMs from the S1/PLGA or S2/PLGA composites may also rely on markedly lower gene expressions of all studied genes except for CXCR4 with the S2/PLGA [29,30,31]. On the other hand, the highest ERK 1/2 activation by the CM from the A1/PLGA may be the overall effect of the increased expressions of VEGF, OPN, and CXCR4 [35,36,37]. Furthermore, the increased expression of c-Fos in cells stimulated by the CM from the A1/PLGA may have contributed to the high ERK 1/2 activity. c-Fos is an early response gene that can be activated due to several physicochemical properties of a material surface, mechanoregulation, as well as local changes in Ca levels [38]. 

Altogether, the present study suggests that the increased hBMSC migration rate for the CM from the A1/PLGA materials may be related to the high amounts of Ca ions released during the first 24 h incubation of the materials in the culture media. This CM from the A1/PLGA not only increased the ERK 1/2 activity but also increased the expression of a set of genes, namely BMP-2, CXCR4, OPN, MMP-2, VEGF, and c-fos. Although this was not explored in this study, all these markers may contribute not only to BMSC migratory activity but also to their osteogenic differentiation. Despite the increased expression of BMP-2 observed for cells treated with the CMs from the S2/PLGA, neither this nor the elevated ERK 1/2 activity were sufficient to increase the migration of these cells; however, a trend of elevated CXCR4 mRNA was noticeable. In contrast, an elevated expression of endogenous BMP-2 in the hBMSCs stimulated by the CM from the A1/PLGA may have sufficiently modulated the expression of CXCR4 [39]. Thus, cells prompted by the CM from the A1/PLGA could have auto-stimulated themselves through the BMP-2 to increase the CXCR4-mediated migration.

Despite that the mechanisms by which the factors analyzed in this work contribute to increased cell motility upon the CM from the A1/PLGA composites are not fully elucidated, we believe it is of importance to report that composite surfaces of specific chemical compositions (i.e., A1/PLGA) display such a biological function. The proposed mechanisms by which these A1/PLGA bioactive composites may simulate cell migration are summarized in Figure 1. Overall, the results of this study may prompt further research to understand the mechanisms by which certain bioactive surfaces/scaffolds attract osteoprogenitor cells. They may also help to further develop the materials that will not only serve as growth surface/scaffolds for cells but also as bioactive stimuli to attract progenitor cells to colonize such growth surfaces/scaffolds, thus accelerating the tissue regeneration process.

## 4. Materials and Methods

### 4.1. Composite Components

The technology of production of the SBG/PLGA composites was reported previously [6,7]. Briefly, PLGA was synthesized via a ring-opening process in the presence of low-toxicity zirconium acetylacetonate as a copolymerization initiator [40]. The molar ratio of l-lactide to glycolide in the copolymer was 85:15, and the molecular masses of PLGA were Mn = 80 kDa and Mw = 152 kDa. Sol-gel glasses used as composites modifiers were of the SiO_2_–CaO (A1—40 mol% SiO_2_, 60 mol% CaO, CaO/SiO_2_ ratio of 1.50; S1—80 mol% SiO_2_, 20 mol% CaO, CaO/SiO_2_ ratio of 0.25) and SiO_2_, CaO, P_2_O_5_ (A2—40 mol% SiO_2_, 54 mol% CaO, 6 mol% P_2_O_5_, CaO/SiO_2_ ratio of 1.35; S2—80 mol% SiO_2_, 16 mol% CaO, 4 mol% P_2_O_5_, CaO/SiO_2_ ratio 0.20) systems, and were added to the PLGA at 50 wt.%. Tetraethoxysilane (TEOS, Si(OC_2_H_5_)_4_, Sigma-Aldrich, St. Louis, MI, USA), triethyl phosphate (TEP, OP(OC_2_H_5_)_3_, Sigma-Aldrich, St. Louis, MI, USA), and calcium nitrate tetrahydrate (Ca(NO_3_)_2_·4H_2_O, POCh, Gliwice, Poland) were used as base components to start the sol-gel process. Then, a 1 M solution of hydrochloric acid (HCl, POCh, Gliwice, Poland) was used as a catalyst for the hydrolysis and condensation reactions. Preformed gels were dried at 40–120 °C for 7 days and then subjected to thermal treatment at 600 °C for 10 h (SiO_2_–CaO), or 700 °C for 20 h (SiO_2_–CaO–P_2_O_5_). Afterwards, they were milled and sieved to obtain bioactive glass powders with an average particle diameter of 45 μm. The physicochemical properties of the SBGs as well as the ion release profile from SBG/PLGA composites have been presented in earlier studies [6,7,41].

### 4.2. Composite Sheets Fabrication

The sheets of the SBG/PLGA composites were fabricated by mixing the SBG particles with a 5% *w/v* PLGA solution in methylene chloride (CH_2_Cl_2_, POCh, Gliwice, Poland) for 24 h, followed by a slip casting of the viscous mixture on glass Petri dishes, the evaporation of the solvent in air, and then drying in air and vacuum to a constant weight. The weight fraction of the SBG in the PLGA was 50 wt.%, similar to recent studies and in contrast to SBG/PLGA sheets containing SBGs at 21 vol% [6,7]. The obtained composite sheets were 0.11 mm thick. 

### 4.3. HBMSC Isolation and Culture Expansion

Unless stated otherwise, all cell culture reagents were purchased from Thermo Fisher Scientific. HBMSCs were harvested from the iliac crest of adult patients (42–67 years old, both genders) according to the approved Institutional Review Board protocol (No. 1072.6120.254.2017). The mononuclear cell fraction was isolated using Ficoll-Paque (GE Healthcare, Chicago, IL, USA) and mononuclear cells were expanded in alpha-Minimum Essential Medium (alpha-MEM) containing 10% mesenchymal stem cell qualified fetal bovine serum (MSCq FBS, Biological Industries) and a 1% mixture of penicillin and streptomycin. Once the primary cultures reached 80–90% confluence, the cells were detached from the bottom of tissue culture flasks with 0.25% trypsin-ethylene-diamine-tetra-acetic acid (EDTA) and either used for the experiments or further expanded in T-75 flasks. All experimental cultures were established with hBMSCs at passages 3–5.

### 4.4. Cultures in the Presence of CMs Harvested from the Materials

To study the effect of ions released from materials on cell response, the material sheets were cut into round disks fitting the bottom of 24-well tissue culture plates. The disks were soaked in 70% ethanol (water solution), washed with phosphate-buffered saline (PBS) to remove ethanol traces, exposed to UV light (10 min each side), and then left overnight under the laminar chamber to dry. The next day, the sterilized materials were soaked for 24 h in 1 mL of growth media to obtain CM. Separately, hBMSCs were seeded on tissue culture plates (TCP) at a density of 2 × 10^4^ cells/cm^2^. After a 24 h culture, the media from above the cells were removed, and the cells were washed with PBS and treated with the CMs harvested from the materials. For the gene expression study, the cells were exposed to the CMs for 72 h. Cells exposed to the CM from PLGA were used as a control. 

### 4.5. Migration Studies

The migration studies were carried out as described earlier [20]. A colorimetric, 24-well (8 µm) QCM Chemotaxis Cell Migration Assay was used to determine the number of cells migrating toward the CM. Before cell seeding into the chambers, they were cultured in serum-free medium for 24 h in order to inhibit cell proliferation. Then, hBMSCs were seeded in the upper chambers in 300 μL of culture medium containing 1 × 10^6^ cells/mL, and the lower chambers were supplemented with 500 μL of the appropriate CMs. The cells were incubated for 24 h at 37 °C. Cells cultured in the presence of the CM harvested from PLGA were used as a control. The cells that had migrated through the membrane were stained according to the manufacturer’s protocol, and the number of cells was determined by reading the absorbance at 650 nm. A schematic performance of the experiments is shown in Figure 3.

### 4.6. Metabolic Activity of Cells

The MTS assay (CellTiter96Aqueous One Solution Cell Proliferation Assay; Promega) was used to determine the cell metabolic activity. After a 3-day culture in the presence of CMs, the cells were washed with sterile PBS and the culture medium from individual wells was replaced with a 0.2 mL solution of 10% MTS reagent in phenol-free alpha-MEM. The plates were incubated at 37 °C until an apparent change of color from yellow to brownish. Then, the colored media were transferred to individual wells in 96-well plates, and the absorbance was recorded at 492 nm using a plate reader (SpectraMax iD3, Molecular Devices, San Jose, CA, USA).

### 4.7. Gene Expression Analyses 

The total RNA was isolated 3-days post-exposure of the cells to the CM. Briefly, TriReagent (MRC Inc.) was used and 0.5 μg of RNA from each culture was reverse-transcribed to cDNA with High-Capacity cDNA Reverse Transcription Kits (Applied Biosystems). The cDNA samples were used for qPCR analyses with TaqMan probes for CXCR4 (Hs00607978), MMP-2 (Hs01548727), and c-Fos (Hs04194186). The reaction mixtures (total volume of 15 μL) for qPCR consisted of 50 ng of cDNA, 7.5 μL of TaqMan Universal PCR Master Mix, and 0.75 μL of TaqMan probe. For qPCR with OPN, BMP-2, VEGF, and TATA primers, each reaction mixture (total volume of 10 μL) contained 50 ng of cDNA, 1 µL of 0.5 µM of gene-specific primers, SYBR^®^Green I, AmpliTaq Gold^®^ DNA Polymerase, and the reaction buffer as recommended by the manufacturer (SYBR^®^ Green PCR Master Mix, Applied Biosystems). All qPCR reactions were carried out in a StepOnePlus Real-Time PCR thermal cycler (Applied Biosystems). The sequences of primers used in the analyses (listed in 5′-3′/forward-reverse order) are as follows: OPN—TGGAAAGCGAGGAGTTGAATG/CATCCAGCTGACTCGTTTCATAA; BMP-2—TGCTAGTAACTTTTGGCCATGATG/GCGTTTCCGCTCTTTGTGTT; VEGF—GAGTGTGTGCCCACTGAGGAGTCCAAC/CTCCTGCCCGGCTCACCGCCTCGGCTT; TATA—GGAGCTGTGATGTGAAGTTTCCTA/CCAGGAAATAACTCTGGCTCATAAC. The PCR reactions for the TaqMan probe were performed for 40 cycles with a denaturation step at 95 °C for 15 s, annealing at 60 °C for 1 min, and elongation at 60 °C for 1 min. For gene-specific primers, the reactions were performed for 40 cycles with a denaturation step at 95 °C for 30 s, annealing at 60 °C for 1 min, and elongation at 72 °C for 30 s. Relative quantification (i.e., the ddCT method) was used to analyze the results with the samples obtained from the cells treated with the CM harvested from PLGA as a reference. 

### 4.8. Western Blot Analyses

The activation of the ERK 1/2 signaling pathway was evaluated 3-days post-exposure of the cells to CM. The cells were washed 3 times with PBS and covered with 200 μL of whole-cell lysis buffer (Cell Signaling Technology, Danvers, MA, USA). The protein concentrations in the extracts were determined with the MicroBCA kit (Pierce). Equal amounts of protein samples were separated on NuPAGE 4–12% Bis-Tris gels under reducing conditions, transferred to polyvinylidene fluoride (PVDF) membranes, and then probed overnight with primary anti-human ERK1/2 and phospho-ERK 1/2 (Cell Signaling Technology, #9102 and #9101, respectively). The horseradish peroxidase-linked secondary antibodies (GE Healthcare) were then applied, and the peroxidase-based signal was detected using Western Lightning Chemiluminescence Reagent Plus (GE Healthcare). The signal was captured on Hyperfilm ECL chemiluminescent films (Perkin-Elmer, Waltham, MA, USA).

### 4.9. Statistical Analyses 

All the data were collected in triplicate and expressed as the mean ± SD. Statistical analyses were performed using Statistica 13 software (Tibco Software, Palo Alto, CA, USA). One-way or multiple ANOVA comparisons with post-hoc Tukey’s test were applied to calculate statistically significant differences at *p* < 0.05.

## Data Availability

The data presented in this study are available on request from the corresponding author.

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
