# Peer review of "Chemical Compounds Released from Specific Osteoinductive Bioactive Materials Stimulate Human Bone Marrow Mesenchymal Stem Cell Migration"

_ijms, 2022, doi:10.3390/ijms23052598_

Round 1

Reviewer 1 Report

This is a potentially interesting manuscript that could contribute to knowledge on the impact of dissolved constituents of bioactive glass-based materials on bone marrow stromal cells.

One questions why the authors didn’t simply use extracts of the four bioactive glasses to carry out this investigation, rather than extracting composites containing the glasses. The absence of this primary information should be explained and accounted for in the manuscript.

The authors must provide the concentrations of dissolved calcium, phosphate and silicate ions in the conditioned media in order to fully interpret the results. The results and discussion sections are underdeveloped because the concentrations of the relevant species in the conditioned media are not given. 

The manuscript is generally clearly written, although requires some editorial corrections to language, grammar and formatting prior to publication. To facilitate the reading of the manuscript, the authors should organise the text in the introduction and discussion into shorter paragraphs according to subject-content.

Are the authors able to provide a more concise and impactful title?

All abbreviated terms, including ‘BMSC’ in the title, and ‘PLGA’ and ‘ERK’ in the abstract should be defined prior to use.

The authors should avoid the use of personal pronouns like ‘we’ and ‘our’ in scientific publications.

Line 16: It would be more informative for the reader if the actual compositions of the four bioactive glasses used in the study were given in the abstract (rather than alluded to in ‘Four SBGs from the SiO2-CaO or SiO2-CaO-P2O5 systems with different CaO/SiO2 ratios…’).

Line 24: State the compositions of the bioactive glasses that significantly enhanced the expressions of the biomarkers.

Lines 48 & 49: Bioactive materials don’t release phosphorus and silicon ions, they release oxyanionic phosphate and silicate species.

Line 53: ‘poly L-lactide-co-glycolide’ should be written, ‘poly(L-lactide-co-glycolide)’.

Line 62: Readers would welcome a more explicit and comprehensive account of the aims and objectives of the study at the end of the introduction to enable them to progress to the results and discussion without having to formally consult the materials and methods section.

Line 75: It is an inconvenience to the reader that the glasses, A1, A2, S1 and S2, are mentioned without the compositions having been given. Readers should not be obliged to search around the manuscript for information that could be easily accessible to them. A more comprehensive account of the aims and objectives at the end of the introduction would resolve this problem.

I have no confidential comments for the editors.

Reviewer 2 Report

This work tested the hypothesis that the ions released from PLGA- sol-gel bioactive glasses (SBG) composites stimulate BMSC migration, by examining a set of biological markers that are involved in the cell migration process, namely the gene expression of c-Fos, OPN, MMP-2, VEGF and CXCR4. A typical cell migration study was also performed.

This work comes into sequence from a previous reported study by the authors (ref 7). In this previous work, authors prepared and characterized a range of composites and evaluated their biological performance. Some of these previous compositions are tested in the present work, namely two from the SiO2–CaO system (A1/PLGA; A2/PLGA) and two from the SiO2–CaO–P2O5 system (S1/PLGA; S2/PLGA). Conditioned media (CM) from the composites were prepared and evaluated for the ability to induce the migration of hBMSC. Authors intended to relate the cell response to the levels of Ca ions present in the CM of the tested composites (A1 > A2 > S1 > S2), referring that this information (ion release profile) was already available in the previous study (ref. 7).

Comments

- The concentrations of Ca, but also of Si and P should be stated in the present work, for the reader information. The reader does not know if there is a big or a little difference on the levels of Ca (and the other ions) among the composites. This might turn to be very relevant to explain the cell response. Further, in the previous work (ref. 7), the number and the nomenclature of the different composites are different, which makes it difficult to collect this information.

- On Mat and Met, the methodology of sections 4.4 and 4.5 should be clarified and detailed: section 4.4 – obtention of the CM - 24-, 6-well plates? Volume – I mL, in both?

Section 4.5 - cell migration studies – Volume of medium in the upper chamber and volume of CM in the lower chamber.

- Results

. A1/PLGA significantly elevated c-Fos, OPN, MMP-2, VEGF and CXCR4 mRNA levels, the transcripts potentially involved in cell migration process, and also the cell migration. This is the only composite that gave consistent results (Figs. 1 and 2). Apparently, due to the higher levels of Ca ions in the CM. Authors provided eventual explanations to justify the results for the gene expression and ERK1/2 activation on this composite.

. The other three composites gave inconsistent results regarding the expression of all genes. The results for S2/PLGA showed a high expression of CXCR4 (a major factor controlling the processes of migration and homing of MSCs) but low levels of c-Fos and VEGF, and OPN (almost negligible). MMP-2 was higher for A1/PLGA (~1.3 vs 1.0 in the control), but all the other composites presented similar values to control. These differences were not attempted to be explained by the authors, may be taking into account the role of the tested genes, i.e. in the earlier migration stage or later, as gene expression was measured after 3 days exposure to the CM (added in cells cultured already for 1 day). Some of the genes (i.e. OPN) have other roles in later stages of cell differentiation. All these issues deserve explanation.

. On essential information that is missing is the metabolic activity of the cells after 3-day exposure to the CM, the time-point used to evaluate the gene expression. Evaluation of ALP (alkaline phosphatase) –, already produced by hBMSC at this stage, would prove the functionality of the cells exposed to the different CMs. The two parameters would assist in eventual explanations of the apparent gene discrepancy results.

. Regarding the ions composition of the CM and the cell response, authors only emphasized Ca ion. The ratio of the ions in the CM does not have any role? Authors mention that, overall, the composites without P2O5 had a better response that those with P2O5. Results on Figs. 1 and 2 do not prove this.

. Why A2/PLGA had a worse performance compared to A1/PLGA? And also compared to the S1 and S2 composites for some genes?

. It is very strange for the reader, that the eventual role Si ion or the different ion ratios were not taken into account in the discussion of the results. 

The concept that the appropriate MSC migration to the material is the essential step for the subsequent cell spreading, proliferation and differentiation is long ago well established, as well that minor material modifications (topography, chemical composition, ion release kinetics, … ) may have significant effects in the cell migration, adhesion and proliferation/differentiation behavior. Thus, from a conceptual point of view, the work has no innovation. In this way, the issues mentioned above should be considered and discussed. Authors should clearly evidence the added contribution for the state of the art.

Round 2

Reviewer 1 Report

The authors have improved the manuscript which requires some editorial corrections of language and grammar prior to publication. 

Reviewer 2 Report

The authors addressed my concerns/sugestions.